SCIENTIFIC CORRESPONDENCE

# Comment on 'Orthogonal lipid sensors identify transbilayer asymmetry of plasma membrane cholesterol'

Kevin C Courtney[1], Karen YY Fung[2], Frederick R Maxfield[3], Gregory D Fairn[2]*, Xiaohui Zha[1,4]*

[1]Department of Biochemistry, Microbiology and Immunology, University of Ottawa, Ontario, Canada; [2]Keenan Research Centre, St. Michael's Hospital, Toronto, Canada; [3]Department of Biochemistry, Weill Cornell Medical College, New York, United States; [4]Chronic Disease Program, Ottawa Hospital Research Institute, Ottawa, Canada

**Abstract** The plasma membrane in mammalian cells is rich in cholesterol, but how the cholesterol is partitioned between the two leaflets of the plasma membrane remains a matter of debate. Recently, Liu et al. used domain 4 (D4) of perfringolysin O as a cholesterol sensor to argue that cholesterol is mostly in the exofacial leaflet (*Liu et al., 2017*). This conclusion was made by interpreting D4 binding in live cells using in vitro calibrations with liposomes. However, liposomes may be unfaithful in mimicking the plasma membrane, as we demonstrate here. Also, D4 binding is highly sensitive to the presence of cytosolic proteins. In addition, we find that a D4 variant, which requires >35 mol% cholesterol to bind to liposomes in vitro, does in fact bind to the cytoplasmic leaflet of the plasma membrane in a cholesterol-dependent manner. Thus, we believe, based on the current evidence, that it is unlikely that there is a significantly higher proportion of cholesterol in the exofacial leaflet of the plasma membrane compared to the cytosolic leaflet.
DOI: https://doi.org/10.7554/eLife.38493.001

*For correspondence:
fairng@smh.ca (GDF);
xzha@ohri.ca (XZ)

**Competing interests:** The authors declare that no competing interests exist.

## Introduction

Cholesterol is an essential molecule in mammalian cells as it supports several critical functions of the plasma membrane and other organelles. The majority of studies have reported that cholesterol constitutes 35–40 mol% of the plasmalemmal lipids (*van Meer et al., 2008*), and most studies support the notion that there is more cholesterol in the cytoplasmic leaflet than in the exofacial leaflet, or that the balance is close to even (*Kobayashi and Menon, 2018*; *Steck and Lange, 2018*). It was surprising, therefore, when Liu et al. reported that the abundance of cholesterol in the exofacial leaflet is about an order of magnitude higher than that in the cytoplasmic leaflet, and that the plasmalemmal cholesterol content is 22–23 mol%. Their conclusions, particularly the cholesterol transbilayer distribution, were based on a series of D4 mutants that require different minimum concentrations, or thresholds, of cholesterol in the membrane to bind liposomes in vitro. However, such thresholds may not depend on cholesterol concentrations alone. Phospholipids that surround cholesterol could influence the accessibility of the D4 probes to cholesterol. Given the complexity of the phospholipid compositions in the plasma membrane and the importance of understanding cholesterol distribution in the plasma membrane, we decided to put these D4 probes, as used by Liu et al., through more rigorous tests. The results here challenge the applicability of this method to quantitatively measure the transbilayer distribution of cholesterol in live cells.

## Results

### Phospholipid head groups impact DAN-D4 binding

For cholesterol-binding, perfringolysin O (PFO) and its derivatives require a minimum cholesterol concentration, or threshold, in the membrane to bind. Such a threshold is known to be strongly influenced by the membrane phospholipid composition (*Flanagan et al., 2009*; *Nelson et al., 2008*; *Sokolov and Radhakrishnan, 2010*). Although Liu et al. stated that D4 binding is unaffected by phospholipid composition, others have reported that it is sensitive to both the acyl chain composition and the phospholipid head group (*He et al., 2017*; *Maekawa and Fairn, 2015a*).

Specifically, Liu et al. used a defined liposome (POPC/POPS/cholesterol) and several D4 variants, labeled with acrylodan (DAN) or NR3, to generate a series of thresholds covering a range of cholesterol concentrations (Figure 1b & c in *Liu et al., 2017*). They then claimed that such thresholds hold true for: (1) DAN-D4 and DAN-D434A for exofacial leaflet mimic (PC/SM) liposomes (Supplementary Figure 1e-h, in *Liu et al., 2017*); and (2) for NR3-YDA and NR3-QYDA for cytoplasmic leaflet mimic (PC/PE/PS/PI) liposomes (Supplementary Figure 1j-k, in *Liu et al., 2017*). These thresholds were then applied to interpret D4 variants' binding to the plasma membrane (Figure 1d & f in *Liu et al., 2017*). To test if this approach is valid, we examined whether DAN-D4 could similarly bind membranes that are as different as the exofacial and cytoplasmic leaflet of the plasma membrane but with identical cholesterol concentrations. For this, we generated liposomes that roughly mimic the exofacial (POPC/egg SM/cholesterol, 36:24:40) or the cytoplasmic leaflets (POPC/POPE/POPS/soy PI/cholesterol, 18:18:18:6:40) of the plasma membrane.

Using identical probes and methodologies to those used by Liu et al., we isolated recombinant D4 from *E. coli*, conjugated the proteins with the solvatochromic dye, DAN, and repeated the in vitro binding experiments as performed by Liu et al. As depicted in *Figure 1A*, for liposomes with constant cholesterol concentration (40%), DAN-D4 preferred the mimic of exofacial leaflet (PC/SM) to the mimic of cytoplasmic leaflet (PC/PE/PS/PI); the amount of D4-DAN that binds the PC/SM liposomes is more than double that which binds PC/PE/PS/PI liposomes. Thus, it is evident that the threshold for D4 is not identical in liposomes that mimic exofacial and cytoplasmic leaflets of the

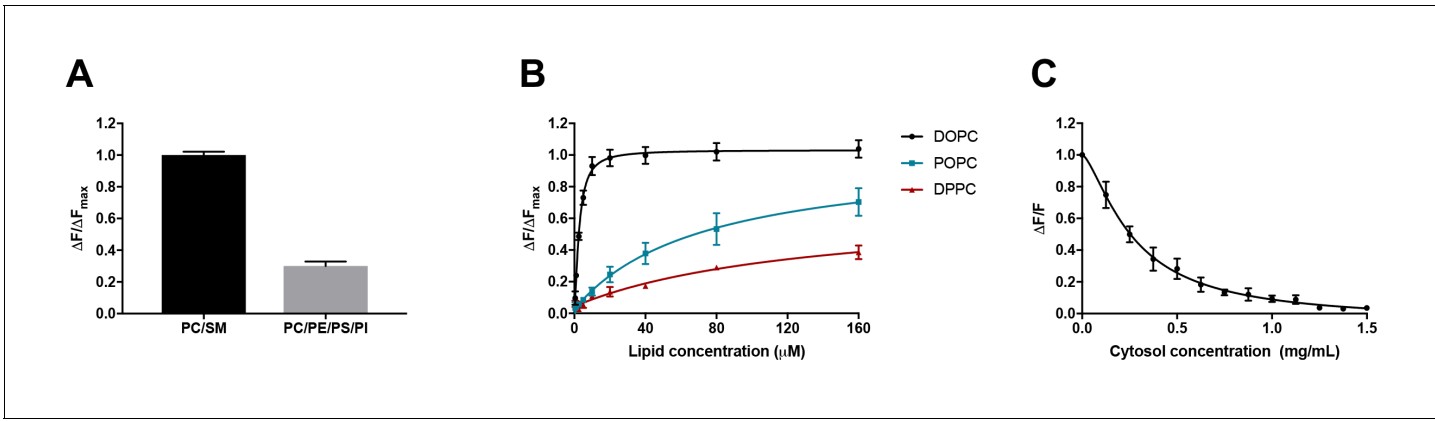

**Figure 1.** D4 binding is influenced by phospholipid composition and is subject to competition from proteins. (**A**) Purified DAN-D4 (0.5 μM) was incubated with 100 μM large unilamellar vesicles (LUVs) composed of POPC/egg SM/cholesterol (36:24:40) and POPC/POPE/POPS/soy PI/cholesterol (18:18:18:6:40). The change in fluorescence emission (ΔF) at 450 nm is used to approximate cholesterol-dependent liposome binding and is corrected for non-specific binding to a cholesterol-free liposome. The results were normalized to the maximal ΔF (ΔFmax). (**B**) DAN-D4 (0.5 μM) binding to increasing concentrations of phosphatidylcholine/cholesterol (60:40) LUVs with various phosphatidylcholine acyl chain saturation. (**C**) DAN-D4 (0.5 μM) binding to 100 μM DOPC/cholesterol (60:40) LUVs in the presence of increasing concentrations of rat liver cytosol. The change in fluorescence was determined relative to cholesterol-free liposomes at 450 nm and then normalized to the control (ΔF/F). All data were acquired with a PTI scanning spectrofluorometer (ex. 380 nm and em. 420–560 nm). Each experiment was repeated at least three times and error bars represent standard error of the mean.

DOI: https://doi.org/10.7554/eLife.38493.002

plasma membrane. Varied DAN-D4 binding in vivo cannot, therefore, be directly interpreted as cholesterol concentration in the plasma membrane. In addition, the phospholipids in the plasma membrane are far more complex than these liposomes, which further challenges the use of these thresholds for calibration as used by Liu et al.

## Acyl chain saturation of phospholipids impacts DAN-D4 binding

It is known that D4 binding to cholesterol in the membrane depends on the degree of cholesterol exposure in that membrane, as a consequence of cholesterol interactions with surrounding phospholipids (*He et al., 2017*; *Maekawa and Fairn, 2015a*). Both head group and acyl chain length impact D4 binding. However, it is the acyl chain saturation, that is, the number of double bonds, that most significantly influences D4 binding (*He et al., 2017*; *Maekawa and Fairn, 2015a*). We therefore sought to test the effect of acyl chain saturation on DAN-D4 binding. To do so, we again used liposomes with constant cholesterol (40 mol%) but varying acyl chain saturation. As depicted in *Figure 1B*, relative to phospholipid with no double bonds in the acyl chain (DPPC, 16:0,16:0), the introduction of a single double bond (POPC, 16:0, 18:1) significantly enhanced the binding of the DAN-D4 to the liposomes. The DAN-D4 binding was further elevated when two double bonds were introduced (DOPC, 18:1, 18:1). Although Liu et al. did study the effects of phospholipid composition on D4 binding (supplementary 1h and i), they did not systematically test the effect of acyl chain saturation in phospholipids, particularly those most abundant in the plasma membrane. Specifically, the lipids used by Liu et al. are primarily POPC and otherwise always contained one saturated and one unsaturated acyl chain. Nevertheless, our experiments clearly demonstrate that both acyl chain saturation and phospholipid head group significantly impact DAN-D4 binding to liposomes. The fact that DAN-D4 binding to the cytoplasmic leaflet-like liposomes was significantly attenuated, compared to those that mimic the exofacial leaflet, suggests that Liu et al. could have underestimated the cholesterol content in the cytoplasmic leaflet of plasma membrane. More importantly, as the lipid classes and species in live cells are significantly more complex than the liposomes we used here, it is unlikely that the liposome-based calibration, as in the approach employed by Liu et al., could be regarded as a true proxy for D4 binding to the exofacial and cytoplasmic leaflets of the plasma membrane. Quantitative interpretation of D4 binding in live cells is extremely difficult, if at all possible, even with multiple rigorous calibrations. Additionally, within cells, the cytoplasm is very rich in proteins, which could further complicate the binding of D4 to cholesterol (see below).

## DAN-D4 binding is highly sensitive to proteins in the medium

Liu et al. reported that microinjected D4 and the variants D4$^{D434A}$ and D4$^{D434A, A463W}$, failed to bind to the cytoplasmic leaflet of the plasma membrane (*Liu et al., 2017*). This was their key evidence to conclude that there is little cholesterol in the cytoplasmic leaflet. This observation was surprising as it has been reported previously that both wild-type D4 and a D434S mutant (comparable to D434A) are capable of binding to the cytoplasmic leaflet of the plasma membrane (*Maekawa and Fairn, 2015b*; *Abe et al., 2012*). This raises the possibility that the microinjected DAN-D4 proteins were not behaving as expected. One of the potential confounders is the presence of cytosolic proteins, which would interfere with DAN-D4 binding to cholesterol in the cytoplasmic leaflet of the plasma membrane. DAN-D4 could bind cytosolic proteins, which would titrate away the microinjected DAN-D4 and prevent DAN-D4 from binding to the cytoplasmic leaflet of the plasma membrane. The protein concentration in cells is estimated to be about 100 mg/ml (*Zeskind et al., 2007*; *Luby-Phelps, 2000*). With such a high protein concentration, even weak affinity of DAN-D4 to cytosolic proteins could reduce the effective concentration available for binding to membranes. To examine the potential impact of proteins, we performed DAN-D4 binding experiments in vitro in the presence of rat liver cytosol (RLC). As depicted in *Figure 1C*, the inclusion of RLC in the binding assay reduced DAN-D4 binding to the liposomes in a dose-dependent manner with an almost complete ablation of binding at 1.5 mg/ml. Thus, the capability of microinjected DAN-D4 to bind the cytoplasmic leaflet of the plasma membrane could be severely diminished in live cells, regardless of cholesterol content in the membrane.

## D4$^{D434A}$ and D4$^{D434A,A463W}$ are capable of binding the cytoplasmic leaflet of the plasma membrane in live cells

mCherry-D4 and mCherry D4$^{D434S}$ have been shown previously to bind to the cytoplasmic leaflet of the plasma membrane. We thus sought to determine whether the new variants of the D4, used by *Liu et al. (2017)*, could similarly bind the cytoplasmic leaflet of the plasma membrane. As shown in *Figure 2A*, exogenously expressed mCherry-D4$^{D434A}$ and mCherry-D4$^{D434A,A463W}$, in fact, do bind to the cytoplasmic leaflet of the plasma membrane. Importantly, these mCherry-tagged probes were responsive to changes in cholesterol: they were displaced from the cytoplasmic leaflet following extraction of the plasmalemmal cholesterol by methyl-β-cyclodextrin (*Figure 2B*). Thus, the lack of binding of the DAN-D4 variants, as shown by Liu and colleagues, is not likely to result from insufficient cholesterol in the cytoplasmic leaflet. Noticeably, liberation of the mCherry-D4 variants from the PM following cholesterol extraction is accompanied by the appearance of bright puncta within the cytosol. This cannot be a result of a sudden increase in endomembrane cholesterol, as acute cholesterol extraction would only lower cellular cholesterol, including endomembranes. However, without cholesterol-rich membrane to bind, D4 could form aggregates within the cytoplasm or be bound to unidentified membrane structures.

## Discussion

It would be significant in membrane biology and physiology if there was a 10-fold enrichment of cholesterol in the exofacial leaflet over that in the cytoplasmic leaflet of the plasma membrane in mammalian cells. In particular, *Liu et al. (2017)*, concluded that generation and maintenance of such a 10-fold gradient would be an ATP-consuming process, incurring a huge energy demand as cholesterol can flip spontaneously in membranes with a $t_{1/2} < 1$ s (*Steck and Lange, 2018*). We sought to interrogate the use of the D4 probes to quantitatively determine their reliability for measuring cholesterol in the exofacial and cytoplasmic leaflets. Here we have repeated and extended vital control experiments and found that the data reported in the study by Liu et al. cannot be extrapolated to provide precise measurements of cholesterol, especially for the cytoplasmic leaflet of the plasma membrane.

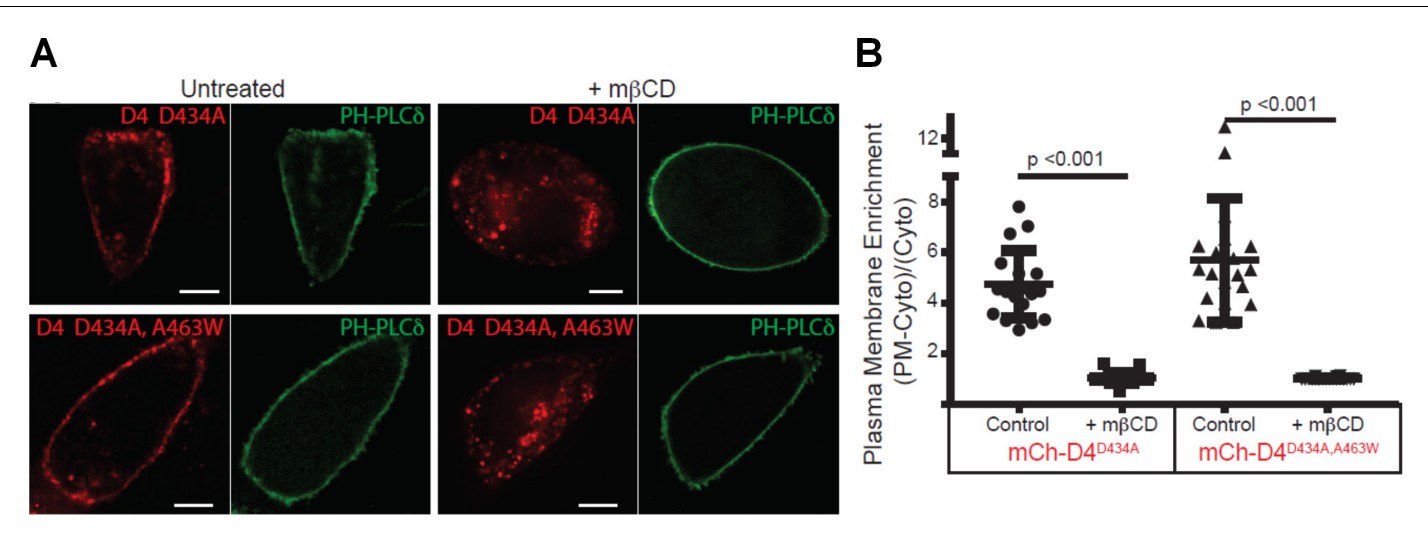

**Figure 2.** D4 variants can bind to the cytoplasmic leaflet of the PM in a cholesterol-dependent manner. (**A**) CHO cells transiently transfected with mCherry-D4$^{D434A}$ and D4$^{D434A, A463W}$ and the plasma membrane marker, Pleckstrin homology domain of phospholipase C δ (PH-PLC δ) were examined using spinning-disc confocal microscopy. (**B**) Live-cell images were acquired of cells expressing the same probes as in (**A**) following incubation with 10 mM methyl-β-cyclodextrin (mβCD) for 20 min to extract plasmalemmal cholesterol. Scale bar, 10 μm. (**C**) Quantitation of the plasmalemmal enrichment of the mCherry signal seen in (**A**) and (**B**). means ± std. dev. n = 20.
DOI: https://doi.org/10.7554/eLife.38493.003

Full-length PFO toxin and recombinant D4 have been studied extensively for their ability to bind cholesterol. It is known that neighboring phospholipids influence the ability of D4 to sense and bind to cholesterol. Because of its small hydroxyl head group, cholesterol has to be shielded from the aqueous environment by surrounding phospholipids. The degree of shielding is influenced by the lateral packing of lipids (i.e., both the acyl chain composition of the lipids as well as the structure of the head group). Thus, when included with lipids with unsaturated acyl chains, such as DOPC, cholesterol is more readily accessible by D4 than when in membranes with saturated DPPC (*Figure 1B*). This observation alone demonstrates that D4 recognizes accessible or chemically active cholesterol, not bulk or total cholesterol.

Still, why does microinjected D4 not bind to the plasma membrane when the heterologous expressed mCherry versions do? One possibility is that the addition of the lipophilic DAN or the Nile red derivative, as used by Liu and colleagues, may increase the affinity of D4 variants to cytosolic proteins. Our observation (*Figure 1C*) supports this possibility. mCherry is not likely to alter the affinity of D4 to cytosolic proteins. Regardless of the fluorophore attached to the D4, the sensors will have to compete with endogenous cholesterol-binding proteins for the accessible pool of cholesterol in membranes.

Genetically coded biosensors for phospholipids have been used by cell biologists for two decades. However, lipid and cholesterol sensors such as D4 must be interpreted with rigor. In particular, cholesterol partitioning between leaflets in the plasma membrane is even more complicated to assess by such binding, as cholesterol can spontaneously flip-flop between leaflets of a bilayer (*Leventis and Silvius, 2001*). Thus, the binding of cholesterol by an exogenous membrane-impermeant probe on the exofacial surface would likely trap cholesterol and alter the cholesterol distribution. This situation was elegantly illustrated in a recent paper, where binding ~1% of the cholesterol in the exofacial leaflet using a D4 homolog was sufficient to trigger a 'lack of cholesterol' signal on the ER membrane inside the cells (*Infante and Radhakrishnan, 2017*). Although D4 and its derivatives are useful in some focused studies to make endpoint measurements or inhibit cholesterol trafficking, we conclude that these tools cannot be used to assess cholesterol partitioning between two leaflets of plasma membranes in live cells.

## Materials and methods

### Recombinant protein production and liposomal binding

Domain 4 (amino acids 391–500) of PFO and its variants were provided by *Liu et al. (2017)*, expressed as GST chimeric proteins using the pGEX-4T-1 vector transformed into BL21 E. coli as previously described. Following purification with glutathione-conjugated affinity resin (GE Healthcare) the recombinant proteins were covalently modified with acrylodan (6-acryloyl-2-dimethylamino-naphthalene) or simply 'DAN' (ThermoFisher) and liberated from the GST tag by incubation with thrombin protease, as described in *Liu et al. (2017)*.

Large unilamellar vesicles were produced by first drying lipids in chloroform in glass tubes under a stream of nitrogen, followed by vacuum dessication for at least 1 hr. The lipids were resuspended in aqueous buffer and subjected to a freeze/thaw cycle before extrusion through 100 nm polycarbonate membrane. The binding of the DAN-D4 to the liposomes results in insertion of DAN into the hydrophobic bilayer that is accompanied by a shift in its emission spectra from a peak of ~490 nm to ~450 nm. The change in fluorescence emission ($\Delta F$) at 450 nm is used to estimate cholesterol-dependent membrane binding, relative to identical but cholesterol-free LUVs. The results were normalized to the maximal $\Delta F$ ($\Delta Fmax$). To examine the effect of a lipid head group, liposomes were generated with the following compositions: exofacial leaflet-like POPC/egg SM/cholesterol (36:24:40) and cytoplasmic leaflet-like and POPC/POPE/POPS/soy PI/cholesterol (18:18:18:6:40). 100 µM liposomes were incubated with 0.5 µM DAN-D4 to determine relative D4 binding. Additionally, to examine the contribution of acyl chain composition, we compared the binding of the D4 to a series of PC/cholesterol (60:40) liposomes in which the molecular species of PC was dipalmitate, palmitate-oleate, or dioleate. DAN-D4 binding curves were generated by incubating 0.5 µM DAN-D4 with increasing liposome concentrations from 0.5 µM to 160 µM. For the effect of rat liver cytosol, DOPC/cholesterol (60:40) liposomes were generated in the same fashion as above but were initially resuspended in diluted rat liver cytosol at the indicated concentration. Again, 100 µM liposomes

were incubated with 0.5 µM DAN-D4 in the presence of increasing rat liver cytosol. The change in fluorescence was determined relative to cholesterol-free liposomes at 450 nm and then normalized to the control (ΔF/F). Fluorescence measurements were aquired with a scanning spectrofluorometer (Photon Technologies International) (ex. 380 nm and em. 420–560 nm).

## Fluorescence microscopy

The open-reading frames of D4$^{D434A}$ and D4$^{D434A, A463W}$ were subcloned into the pmCherry-C1 expression plasmid. Chinese hamster ovary (CHO) cells were maintained in DMEM media containing 10% fetal bovine serum. For imaging experiments, cells were seeded on 18 mm coverslips and transiently transfected with the indicated plasmids using X-tremeGENE9 (Roche) and returned to the incubator. The next day, live cells were transferred to a chamber slide and imaged using spinning-disc confocal microscopy. To determine the impact of cholesterol removal on D4 localization, cells were treated with 10 mM methyl-β-cyclodextrin for 20 min before imaging. The spinning-disc imaging system used is based on a Leica DMIRE2 equipped with a Yokogawa CSU X1 scan head and a 60 × (NA 1.35) oil immersion objective using a Hamamatsu C9100-13 electron-multiplying charge-coupled device (EM-CCD) camera. Excitation light was provided by 491 nm (50 mW) and 561 nm (50 mW) lasers, and emitted light was collected after passage through 515/40 and 594/40 nm emission filters. Post-acquisition analysis was conducted using the region of interest tool in ImageJ. Briefly, in highly magnified images regions of the plasma membrane (PM), cytosol, and outside the cell (background) were analyzed for mean fluorescence intensity. The plasma membrane enrichment was calculated using the background subtracted values and the following equation; (PM-Cyto) ÷ Cyto. The graph was generated using Prism (GraphPad) and includes the individual data points (n = 20) with the means ± the standard deviation indicated.

## Acknowledgement

We would like to acknowledge the valuable insight and feedback that we received from Theodore L. Steck and Yvonne Lange throughout the development of this work. This work was supported by MOP-130453, Canadian Institutes of Health Research (CIHR) and RGPIN 40210–2013, Natural Sciences and Engineering Research Council of Canada (NSERC) (XZ); R01 GM123462 (NIH) (FRM), and MOP-133656, Canadian Institutes of Health Research (CIHR) (GF).

## Additional information

### Funding

| Funder | Grant reference number | Author |
|---|---|---|
| Canadian Institutes of Health Research | Operating Grant MOP-130453 | Xiaohui Zha |
| Natural Sciences and Engineering Research Council of Canada | Discovery Grant RGPIN 40210-2013 | Xiaohui Zha |
| National Institutes of Health | R01 GM123462 | Frederick R Maxfield |
| Canadian Institutes of Health Research | MOP-133656 | Gregory D Fairn |

The authors declare that there was no funding for this work

### Author contributions

Kevin C Courtney, Conceptualization, Data curation, Formal analysis, Validation, Investigation, Visualization, Methodology, Writing—original draft, Writing—review and editing; Karen YY Fung, Validation, Investigation, Visualization, Methodology, Performed substantial amount of mCherry experiments, Analyzed data; Frederick R Maxfield, Conceptualization, Resources, Data curation, Supervision, Methodology, Writing—original draft, Writing—review and editing; Gregory D Fairn, Resources, Data curation, Supervision, Investigation, Methodology, Writing—original draft, Writing—review and editing; Xiaohui Zha, Conceptualization, Resources, Data curation, Formal analysis,

Supervision, Funding acquisition, Validation, Investigation, Visualization, Methodology, Writing—original draft, Project administration, Writing—review and editing

### Author ORCIDs
Kevin C Courtney (iD) http://orcid.org/0000-0003-1315-4917
Frederick R Maxfield (iD) http://orcid.org/0000-0003-4396-8866
Xiaohui Zha (iD) http://orcid.org/0000-0003-2873-3073

### Decision letter and Author response
Decision letter https://doi.org/10.7554/eLife.38493.007
Author response https://doi.org/10.7554/eLife.38493.008

## Additional files

### Supplementary files
• Transparent reporting form
DOI: https://doi.org/10.7554/eLife.38493.004

### Data availability
All data generated or analysed during this study are included in the manuscript.

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
