## [Decision Letter]

Thank you for submitting your article "Comment on "Orthogonal lipid sensors identify transbilayer asymmetry of plasma membrane cholesterol"” for consideration by *eLife*. Your article has been reviewed by Philip Cole as the Senior Editor, a Reviewing Editor, and three reviewers. The following individuals involved in review of your submission have agreed to reveal their identity: Satyajit Mayor (Reviewer #1); Daniel Wüstner (Reviewer #3). A further reviewer remains anonymous.

The reviewers have discussed the reviews with one another and the Reviewing Editor has drafted this decision to help you prepare a revised submission.

Summary:

The authors challenge the conclusions made in a recently published paper regarding the distribution of cholesterol between the two leaflets of plasma membranes (PMs) (Liu et al., 2017). Using a domain from a cholesterol-binding bacterial protein, this earlier paper had reported that the outer leaflet of the PM is ~12-fold enriched in cholesterol compared to the inner leaflet, a finding that is at odds with many published studies that report a roughly even distribution of cholesterol between the two PM leaflets. The current "Comment" submission raises several valid concerns regarding the sensor used by Liu et al., and how that may have skewed the analysis in their study. The subject of trans-bilayer cholesterol distribution has received renewed attention in light of its significance in understanding cellular cholesterol transport (mediated by NPC1 and other proteins) and cholesterol-mediated cellular signaling by the Hedgehog signaling pathway. The reviewers agreed that this "Comment" challenging the methodology and analysis used by Liu et al., would serve not just as a cautionary critique but provide a timely and clarifying contribution to this field.

Essential revisions:

To make this Comment article as solid as possible, the reviewers request the following essential revisions:

1) The authors show the effect of composition on DAN-D4 binding at a single concentration of D4 (which incidentally is not mentioned in the text). It would be desirable that a binding isotherm be performed/shown that indicates that they are achieving at saturation levels for all compositions tested. It is not clear whether it is the 'availability/accessibility' of cholesterol that is changing or the Kd of D4 for the cholesterol in the membrane. Coupled with the lack of information about the concentrations of the D4 protein, it is hard to come to a firm conclusion.

2) The authors challenge the lack of binding of the DAN- and NR3-labeled D4s to the inner PM leaflet in the Liu et al. study by showing that heterologously expressed mCherry D4s do label the inner leaflet. The impact of this new data piece is modest: transiently transfected fluorescent fusions of D4 probes have been previously reported to localize to the inner plasma membrane leaflet and with the substantial cell-to-cell variation in the labeling pattern observed with transient expression, single cell exemplary images are questionable. The authors suggest that lack of detection of the NR3-D4 at the inner leaflet may be due to low amounts of D4s being microinjected into cells or due to the presence of competing endogenous proteins (the authors use BSA as a case study to make this point). The authors' discussion of this point is not compelling since transient transfection may be even more difficult to control than microinjection in terms of protein dosage and it is not apparent why the transiently expressed protein would behave more correctly. This should be quantitatively addressed by comparing their binding isotherms – it is possible that mCherry D4 variants behave differently from both NR3-D4 variants or the DAN D4 variants.

3) The authors need to be cautious in their interpretations regarding point 2 – the exoplasmic leaflet binding may also be subject to interference by proteins.

4) Liu et al., did test liposomes with substantial amounts of a typical inner leaflet lipid (20 mol% PS, in addition to PC and cholesterol) and also PC/PE/PS/PI containing liposomes, but not systematically with all D4 variants. The authors here are more systematic and additionally show that the acyl chain composition of PC affects D4 binding. They make a valid point but should not mischaracterize Liu et al.

[Editors' note: further revisions were requested prior to acceptance, as described below.]

Thank you for resubmitting your work entitled "Comment on "Orthogonal lipid sensors identify transbilayer asymmetry of plasma membrane cholesterol”" for further consideration at *eLife*. Your revised article has been favorably evaluated by a Senior Editor (Philip Cole) and a Reviewing Editor (Arun Radhakrishnan).

The manuscript has been improved but there are some remaining issues that need to be addressed before acceptance, as outlined below:

Please make the following clarifying revisions.

1) Results section: Please include an additional reference to the large body of earlier work that also suggests ~100 mg/ml protein concentration in cytosol, for instance: "Cytoarchitecture and physical properties of cytoplasm: volume, viscosity, diffusion, intracellular surface area." Luby-Phelps, 2000.

2) Results section: Figure 2 clearly shows labeling of the inner leaflet of the PM under normal "untreated" conditions, which disappears upon cholesterol depletion. However, the intense bright puncta observed upon cholesterol depletion may confuse the readers and lead them to think that cholesterol content of an internal organelle like the lysosome increases upon cholesterol depletion! We assume that the authors do not think this is the case. The observed staining is most likely D4 aggregates in cytosol, since the D4 has to go somewhere once membrane cholesterol is depleted. A sentence clarifying this point would be useful.

---

## [Author Response]

Essential revisions:To make this Comment article as solid as possible, the reviewers request the following essential revisions:1) The authors show the effect of composition on DAN-D4 binding at a single concentration of D4 (which incidentally is not mentioned in the text). It would be desirable that a binding isotherm be performed/shown that indicates that they are achieving at saturation levels for all compositions tested. It is not clear whether it is the 'availability/accessibility' of cholesterol that is changing or the Kd of D4 for the cholesterol in the membrane. Coupled with the lack of information about the concentrations of the D4 protein, it is hard to come to a firm conclusion.

We now performed DAN-D4 binding curves, particularly for PC with different acyl chain unsaturation (Figure 1B), which is well-documented as the most significant factor influencing cholesterol accessibility. The result is consistent with the conclusion in previous version, and also with existing literature, that D4 accessibility of cholesterol is increased with increasing acyl chain unsaturation. Accessibility will for sure directly affect Kd for cholesterol: the more accessible the cholesterol, the lower the Kd for cholesterol. We apologize for the omission of the details. They are now added in the Materials and methods section.

2) The authors challenge the lack of binding of the DAN- and NR3-labeled D4s to the inner PM leaflet in the Liu et al. study by showing that heterologously expressed mCherry D4s do label the inner leaflet. The impact of this new data piece is modest: transiently transfected fluorescent fusions of D4 probes have been previously reported to localize to the inner plasma membrane leaflet and with the substantial cell-to-cell variation in the labeling pattern observed with transient expression, single cell exemplary images are questionable. The authors suggest that lack of detection of the NR3-D4 at the inner leaflet may be due to low amounts of D4s being microinjected into cells or due to the presence of competing endogenous proteins (the authors use BSA as a case study to make this point). The authors' discussion of this point is not compelling since transient transfection may be even more difficult to control than microinjection in terms of protein dosage and it is not apparent why the transiently expressed protein would behave more correctly. This should be quantitatively addressed by comparing their binding isotherms – it is possible that mCherry D4 variants behave differently from both NR3-D4 variants or the DAN D4 variants.

We agree with the reviewers that the previous explanation we provided as to why the DAN-D4s don’t bind to the PM is rather unsatisfying. In the revised manuscript, we now performed DAN-D4 binding competition with rat liver cytosol (Figure 1C). Result shows that DAN-D4 binding to membrane is highly sensitive to cytosol proteins. Our results suggest that the addition of the lipophilic DAN molecule may increase binding to unknown cytosolic proteins. The mCherry-labeled D4 variants are not likely subjected to such interference. Indeed, a large amount of literate reported that the localization of a different variant, mCherry-D4^D434S^ and its responsiveness to cholesterol extraction, oxidation, supplementation and mislocalization. (see PMID: 25663704, 26572827, 28564600, 28391244 and others for examples).

Nevertheless, in the revised manuscript, we provide additional quantitation and ratiometric analysis for the mCherry-D4 D434A and D434A/A463W in control cells and following 15 min of treatment with 10 mM methyl-b-cyclodextrin. In our experience, using spinning-disc confocal microscopy with live cells we see little heterogeneity with these constructs.

3) The authors need to be cautious in their interpretations regarding point 2 – the exoplasmic leaflet binding may also be subject to interference by proteins.

We appreciate the concern of the reviewers. However, it is worth pointing out that the protein concentration in the cytoplasm is at least 10-fold greater than that in the extracellular medium used for in vitrocell culture experiments. To improve the readability of the manuscript we clarified this point in the text.

4) Liu et al., did test liposomes with substantial amounts of a typical inner leaflet lipid (20 mol% PS, in addition to PC and cholesterol) and also PC/PE/PS/PI containing liposomes, but not systematically with all D4 variants. The authors here are more systematic and additionally show that the acyl chain composition of PC affects D4 binding. They make a valid point but should not mischaracterize Liu et al.

Thank you for the comment, it is well taken, and our language was too strong. In the revised version of the manuscript we have re-written that section. Ultimately, the point that we were trying to make initially remains valid. The binding of D4 to membranes varies significantly, not only between liposomes that mimic the outer vs. inner leaflets, but even more strikingly when acyl chain composition of the liposomes varied. Therefore, it is not appropriate to use these simplified liposomes to as standard curve for interpolation of data obtained from cell-based assays.

[Editors' note: further revisions were requested prior to acceptance, as described below.]

The manuscript has been improved but there are some remaining issues that need to be addressed before acceptance, as outlined below:Please make the following clarifying revisions.1) Results section: Please include an additional reference to the large body of earlier work that also suggests ~100 mg/ml protein concentration in cytosol, for instance: "Cytoarchitecture and physical properties of cytoplasm: volume, viscosity, diffusion, intracellular surface area." Luby-Phelps, 2000.

Referenced added.

2) Results section: Figure 2 clearly shows labeling of the inner leaflet of the PM under normal "untreated" conditions, which disappears upon cholesterol depletion. However, the intense bright puncta observed upon cholesterol depletion may confuse the readers and lead them to think that cholesterol content of an internal organelle like the lysosome increases upon cholesterol depletion! We assume that the authors do not think this is the case. The observed staining is most likely D4 aggregates in cytosol, since the D4 has to go somewhere once membrane cholesterol is depleted. A sentence clarifying this point would be useful.

Following sentences are added:

“Noticeably, the liberation of the mCherry-D4 variants from the PM following cholesterol extraction is accompanied by the appearance of bright puncta within the cytosol. This could not be due to a sudden increase in endomembrane cholesterol, as acute cholesterol extraction would only lower cellular cholesterol, including endomembranes. However, without cholesterol-rich membrane to bind, D4 could form aggregates within the cytoplasm or be bound to unidentified membrane structures.”